# Cryptic coral community composition across environmental gradients

Gia N. Cabacungan[1], Tharani N. Waduwara Kankanamalage[1], Amilah F. Azam[1], Madeleine R. Collins[1], Abigail R. Arratia[1], Alexandra N. Gutting[2], Mikhail V. Matz[1], Kristina L. Black[1]*

1 Department of Integrative Biology, University of Texas at Austin, Austin, Texas, United States of America,
2 The Nature Conservancy, St. Croix, U.S. Virgin Islands, United States of America

* kblack@utexas.edu

## Abstract

Cryptic genetic variation is increasingly being identified in numerous coral species, with prior research indicating that different cryptic genetic lineages can exhibit varied responses to environmental changes. This suggests a potential link between cryptic coral lineages and local environmental conditions. In this study, we investigate how communities of cryptic coral lineages vary along environmental gradients. We began by identifying cryptic genetic lineages within six coral species sampled around St. Croix, USVI based on 2b-RAD sequencing data. We then analyzed associations between the distributions of cryptic lineages across the six coral species (i.e., "cryptic coral community composition") and ecoregions, or geographically distinct environmental conditions. Our findings show that depth is a more significant predictor of community composition than ecoregions and is the most influential factor among the 40 abiotic variables that characterize ecoregions. These results imply that cryptic coral communities are influenced by both depth and local environmental conditions, although the exact environmental factors driving these patterns remain unknown. Understanding community turnover across a seascape is important to consider when outplanting corals to restore a reef, as locally-adapted lineages may have differential fitness in different environmental conditions.

## Introduction

The Caribbean Sea is experiencing a sharp decline in coral cover, primarily due to diseases [1, 2] and warm temperature anomalies [3–5]. These stressors have led to rapid transformations in coral communities and a subsequent reduction in reef functionality across the Caribbean [6]. In response, conservationists have launched efforts to restore coral populations by outplanting asexually propagated coral fragments [7] and sexually propagated coral recruits [8] onto degraded reefs. Restoration activities are widespread, covering multiple Caribbean islands [9, 10], the U.S. [11], and Central and South America [12]. When outplanting coral, restoration practitioners aim to identify specific genotypes that demonstrate resilience, i.e., high survival and growth rates, at outplant sites. However, it's important to consider that the

**Data Availability Statement:** All sequences and metadata are deposited on the Sequence Read Archive under Bioproject PRJNA1122865. Tutorials

for these analyses, as well as all scripts and data to reproduce the figures in this manuscript, can be found at: https://github.com/kristinaleilani/2bRAD-workshop.

**Funding:** This study was primarily funded by the NatureNet Science Fellowship from The Nature Conservancy to K. L. B., and partially funded by the National Fish and Wildlife Foundation Grant 0318.20.069532 awarded to The Nature Conservancy and the National Science Foundation grant OCE-2433977 to M.V.M.. However, the funders had no role in study design, data collection and analysis, decision to publish, or preparation of the manuscript."

**Competing interests:** The authors have declared that no competing interests exist.

most successful genotypes may vary due to local environments at different outplanting locations [13].

Understanding how environmental gradients drive local adaptations will be crucial for optimizing outplanting strategies. By ensuring corals are placed in environments where they are most likely to thrive, we can enhance their survival rates [13]. It is also important to recognize shared environmental adaptations across various coral species. These species likely form communities that are locally adapted to specific, yet undefined, ecoregions [14]. Thus, improving the survival rates of outplanted corals necessitates a robust understanding of how local environmental factors shape the genetic structure of diverse coral species. This knowledge is vital for developing targeted, effective restoration strategies that support the resilience and recovery of coral ecosystems.

Many coral taxa around the world, including *Acropora hyacinthus* [15], *Orbicella faveolata* [16], *Pocillopora spp.* [17], *Porites cf. lobata* [18], and *Agaricia spp.* [19], exhibit cryptic genetic variation, forming distinct genetic lineages that are morphologically similar but genetically divergent. Genetic divergence within species can arise from various factors- including differences in depth, habitat type, physical disturbances, oceanographic factors, temperature, or geographic isolation [20]. Depth is a common driver in many species [20], with numerous studies demonstrating significant associations between cryptic lineages and depth [17, 20, 21]. Depth preferences often lead to spatial segregation; for instance, cryptic genetic lineages of *M. cavernosa* and *S. siderea*, are found at different depths and distances from shore, with some lineages restricted to shallow reefs of 3–10 meters, while others thrive in deeper habitats over 20 meters [21]. Similarly, *Agaricia fragilis* exhibits genomic divergence and limited dispersal across depths [22].

In addition to depth, environmental factors such as salinity [23, 24] and temperature [25, 26] significantly influence community composition in marine ecosystems. Temperature is of particular concern due to increasing heat anomalies leading to mass mortality events [3–5]. Some cryptic lineages exhibit variable responses to temperature [15, 16]. For example, cryptic lineages of *O. faveolata* in Panama show different physiological responses to coral bleaching, suggesting adaptations to rising water temperatures [16]. Likewise, cryptic lineages of *A. hyacinthus* in American Samoa demonstrate variations in heat tolerance and their association with heat-resistant symbionts [15]. More broadly, shifts in community composition can result from divergent responses to heat stress [27], although communities may recover if environmental conditions reverse [28]. Given the consistent associations of depth and temperature with cryptic genetic variation in coral species, we hypothesize that these factors may serve as key environmental drivers of cryptic coral community composition.

St. Croix is characterized by extensive patch reefs and colonized pavement areas, and it includes protected areas such as the Buck Island Reef National Monument and the East End Marine Park [2] (Fig 1). Despite facing the widespread environmental pressures that have decimated coral populations elsewhere, the benthic habitats of St. Croix have historically maintained fair conditions, with some regions like Buck Island Reef showing notable resilience [29, 30]. This resilience is often attributed to the genetic diversity among coral species, with particularly resilient species like *M. cavernosa* and *P. astreoides* playing a crucial role in recovery potential [29, 31]. Alternatively, the *Orbicella* populations on the island, especially *O. faveolata*, have suffered significant declines, characterized by dead colony surface areas and erosion rates that surpass their live surface area and rates of calcification [32]. Furthermore, the recent surge in Stony Coral Tissue Loss Disease (SCTLD) throughout the U.S. Virgin Islands underscores the urgency of evaluating the variability in resilience within coral communities [33].

In this study, we conducted a population genomics analysis of six coral species sampled across St. Croix: *Agaricia agaricites*, *Montastraea cavernosa*, *Orbicella faveolata*, *Porites*

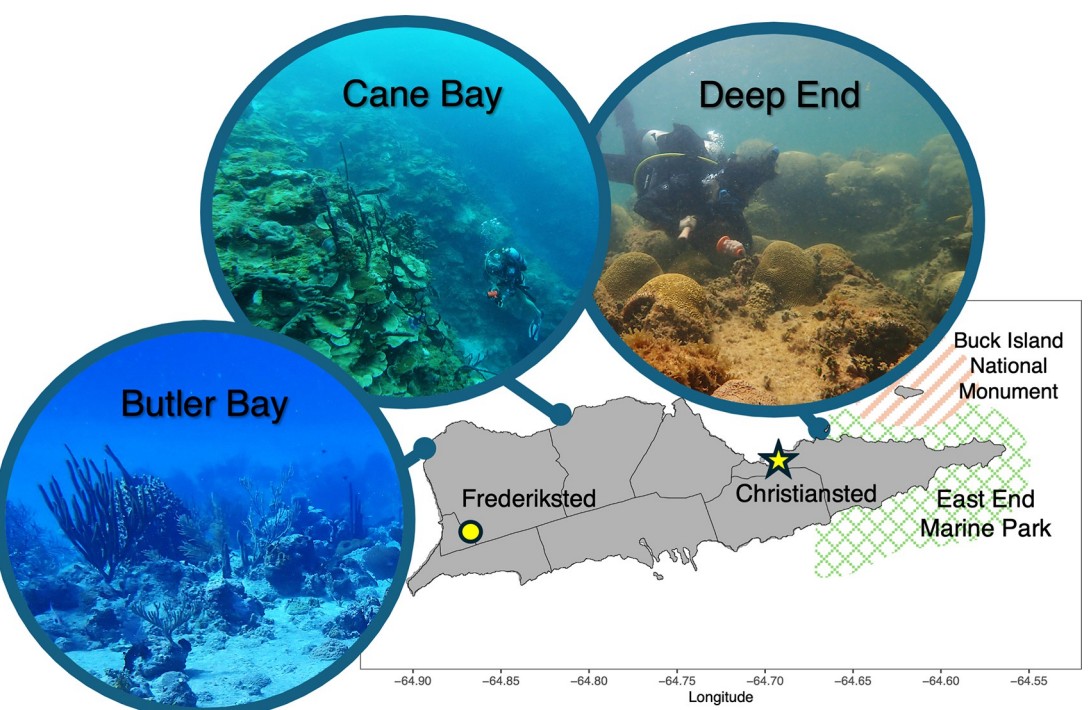

**Fig 1. Map of St. Croix, USVI.** Three coral sampling sites are pictured to demonstrate the variability of reef environments across the island. On the western side, Butler Bay (photo by Corina Marks) has high visibility with a moderate reef slope. Cane Bay (photo by Kristina Black) is most notably defined by a "wall," or a sudden cavernous drop-off adjacent to the north shore. Deep End (photo by Daisy Flores) is shallow and turbid. Some sites receive direct anthropogenic impacts from the capital Christiansted, which is highly developed, and the town of Frederiksted, where cruise ships bring tourism. The island also maintains two marine protected areas: Buck Island National Monument and East End Marine Park. Coastlines and district boundaries are plotted with GADM mapping data (https://gadm.org/index.html).

*astreoides*, *Pseudodiploria strigosa*, and *Siderastrea siderea*. These species, which are wide-spread throughout the Caribbean, represent a diverse range of coral families and reproductive strategies, including both broadcast spawners and brooders. Our sample set also included a mix of species experiencing population declines, such as the endangered *O. faveolata* [34], as well as species that are increasing and considered "weedy," such as *P. astreoides* [35, 36].

For each species, we identified cryptic genetic lineages and subsequently explored the environmental associations between seascape heterogeneity and the composition of these cryptic coral communities. The primary goal of our research was to pinpoint the environmental predictors and ecological thresholds that play a role in structuring the distribution of cryptic genetic lineages across multiple species. This comprehensive approach allows us to better understand the dynamics influencing coral diversity and resilience across different environmental gradients.

## Methods

### Sample collection

Six coral species- *A. agaricites* (n = 42), *M. cavernosa* (n = 54), *O. faveolata* (n = 98), *P. astreoides* (n = 61), *P. strigosa* (n = 108), *and S. siderea* (n = 53)—were sampled from 12 sites surrounding St. Croix. Adult colonies >5cm in diameter were sampled, and for each sample, depth was recorded *in situ* (S1 Fig in S1 File). Tissue samples were stored in 100% ethanol at -80 degrees Celsius. Sampling was conducted in accordance with the U.S. Virgin Islands

Department of Planning and Natural Resources (The Nature Conservancy Coral Restoration Permit DFW20052X).

## Laboratory methods

Genomic DNA was isolated using a CTAB extraction procedure (Supplementary section 1 in S1 File), followed by purification using the Zymo Genomic DNA Clean & Concentrator kit (Zymo #D4067) following the manufacturer's protocol. All samples were equalized to 12ng/µL, and 2bRAD libraries were prepared following a protocol available at https://github.com/z0on/2bRAD_denovo. 2bRAD is a restriction site-associated DNA sequencing method used to survey 0.5% of the total genome, which is sufficient for profiling neutral genetic variation of these natural populations [37]. The libraries were sequenced at the Genomic and Sequencing Analysis Facility at the University of Texas at Austin on the Illumina NovaSeq SR100 platform.

## 2bRAD genotyping

Raw sequences were processed using the Texas Advanced Computing Center (TACC). Raw reads were trimmed and deduplicated following a custom pipeline hosted at https://github.com/z0on/2bRAD_denovo, then low-quality ends were trimmed using Cutadapt [38]. Reference genomes were available for mapping sample sets of *O. faveolata* (NCBI RefSeq assembly: GCF_002042975.1) and *M. cavernosa* [21]. However, for all other species, *de novo* cluster-derived reference was constructed, following [37]. Briefly, the trimmed reads within each sample set were "stacked" to identify tags that appear multiple times. Tags that appeared in at least 10 individuals were collected. Then, tags with more than 7 observations without reverse-complement were discarded, and the remaining tags were clustered at 91% identity (i.e. allowing for up to 3 mismatches within 34b tag). The most abundant tag from each cluster became the reference, and all the reference tags were concatenated to form 10 equal-sized pseudo-chromosomes. Additionally, four reference genomes of the main zooxanthellae clades of algal symbiont genomes were concatenated onto the coral genomes (*Symbiodinium*: NCBI accession no. GCA_003297005.1, *Brevolium*: GCA_000507305.1; *Cladocopium*: GCA_003297045.1; *Durisdinium*: GAFP00000000). All genomes were indexed with Bowtie2 [39] and trimmed reads were mapped to their respective reference. All reads that mapped to the symbiont genomes were discarded, leaving only coral reads for downstream analysis. The resulting bam files were genotyped with ANGSD [40] and individuals with less than 10% of sites at 5X coverage were discarded. Sites were filtered with minor allele frequency $< 0.025$, and only sites with mapping error $<0.1\%$ and genotyped in at least 75% of individuals were retained. Genotypes were compiled into a pairwise Identity-by-state (IBS) genetic dissimilarity matrix for initial inspection. Hierarchical clustering of the genetic matrix was evaluated for correct alignment of technical replicates, and identification of clonal samples. All clones and technical replicates were removed. Then the samples were re-genotyped with ANGSD using the smaller set of individuals to produce the final IBS matrix.

## Population genomics

We explored population structure within each species by visualizing the IBS dissimilarities as a hierarchical clustering tree and principal coordinates analysis (PCoA) using the R package *vegan* (version 2.6–4). For *O. faveolata* and *P. strigosa*, the optimal number of genetic lineages was visually determined by examining hierarchical clustering tree and PCoA. To determine the number of lineages in the remaining four species, we clustered our samples with larger 2bRAD datasets from the Florida Keys and Gulf of Mexico. Four cryptic genetic lineages in *M. cavernosa* and *S. siderea* were detected in a previous study [21, 41] NCBI BioProject Accession

PRJNA679067), and at least two more lineages of *M. cavernosa* were detected in the Gulf of Mexico [42]. Three lineages of *A. agaricia* and *P. astreoides* were detected in another study from Florida [43] SRA Bioproject PRJNA812916). *M. cavernosa*, *S. siderea*, and *A. agaricites* all demonstrated clear assignment to previously detected lineages. However, *P. astreoides* did not cluster with any lineages from Florida, despite genetic connectivity previously detected between Florida and the U.S. Virgin Islands in a microsatellite study [44]. Therefore, we also designated the optimal number of lineages within St. Croix *P. astreoides* by visual separation in a hierarchical clustering tree and principal coordinate analysis (PCoA).

## Defining ecoregions

Environmental monitoring data was obtained from the Virgin Islands Department of Planning and Natural Resources (DPNR), available at waterqualitydata.us. This data represents eight *in situ* variables measured across the St. Croix coastline from 2000 to 2022 (S2 Fig and S1 Table in S1 File). Variables included pH, *Enterococcus* and *E. coli* (count per 100ml), dissolved oxygen, Kjeldahl nitrogen, and phosphorus (mg/L), and Secchi disk depth (m). All variables were monitored across all coastlines of St. Croix, and very close to coral sampling sites (S2 Fig in S1 File). The only exception is that E. coli was not monitored near Cane Bay on the north shore. We summarized variables at each location by calculating mean, maximum, and minimum values across all observations. We also calculated mean monthly range as the difference between the maximum and minimum value at each site each month, and then averaged across months (S2 Table in S1 File). Similarly, we calculated mean yearly range as range of values at each site recorded over each year, averaged across years (S3 Table in S1 File). Altogether, these measures produced 40 environmental variables total. To extract environmental values at our coral sampling sites, we performed a kriging interpolation of each variable. Using the autoKrige function in the R package *automap* (version 1.0–14), we inferred the optimal model to fit a variogram between neighboring monitoring sites and implemented cross-validation to approximate a continuous environmental grid. Then, values were extracted at the twelve coral sampling sites from environmental grids of each variable (S3 Fig in S1 File).

Due to multicollinearity within our environmental dataset, we aimed to summarize conditions across the seascape by clustering sampling sites into distinct "ecoregions" of similar environmental values. We first reduced multicollinearity within the environmental dataset by removing variables with height < 0.3 average distance in a hierarchical clustering tree (Fig 2A). Then the smaller subset of 15 variables was used to cluster sampling sites by environment (Fig 2B). The hierarchical clustering tree resulted in four ecoregions (Fig 2C) across the seascape. At first glance, these regions appear to align with visual differences between the reef environments (Fig 1). These regions are also reasonable given the influence of the Caribbean Current around St. Croix. The northbound Caribbean current deflects around the southern shore of St. Croix, sending disparate wake flows to the eastern sites (ecoregion A) and the western sites (ecoregions C and D) (45). The two flows are asymmetric due to wake eddies, or circular currents, forming contained benthic conditions in ecoregion A [45]. Ecoregion B (The Palms and WAPA) are closest to the Virgin Islands capital of Christiansted, and likely receive land-based sources of pollution [46]. Ecoregion C (Cane Bay, North Star, and Carambola) is close to ecoregion D in the hierarchical clustering tree (Fig 2B) but likely differs due a prominent shelf break that drops 5,500m deep approximately 250m from shore [47]. Together, these ecoregions likely capture broad environmental heterogeneity across the seascape, so that even variables that are missing from this analysis are likely congruent with these environmental boundaries.

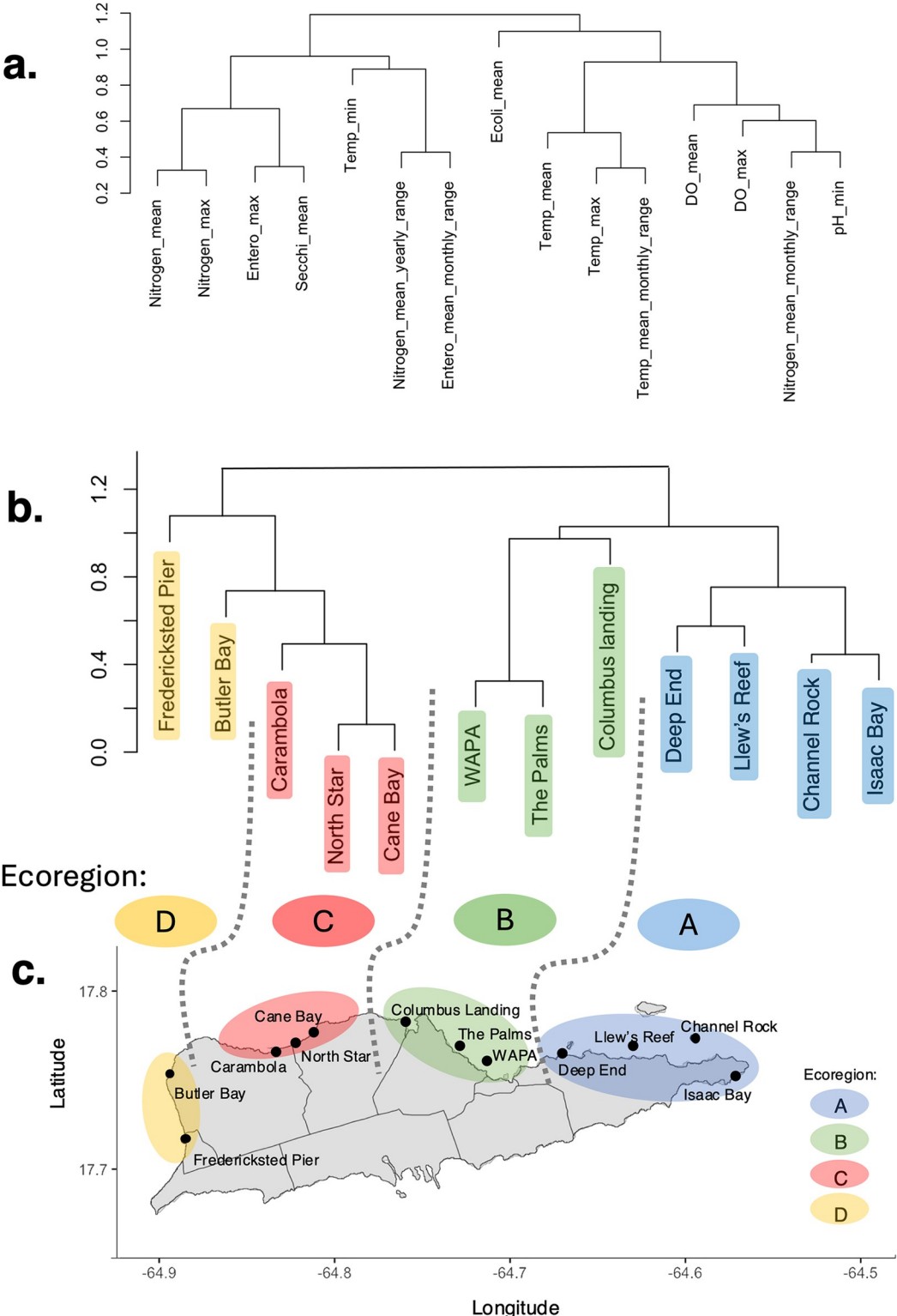

**Fig 2. Environmental clustering of 12 sites into "ecoregions".** (a) Clustering of 16 environmental variables, after reducing multicollinearity. (b) Clustering of sampling sites based on those 15 environmental variables. Ecoregion is indicated by color and assigned to sites that cluster by environment. (c) Sampling sites and their ecoregion assignment on the map of St. Croix, USVI. Coastlines and district boundaries are plotted with GADM mapping data (https://gadm.org/index.html).

## Community-environment associations

A common framework for finding associations between community composition and environmental variables is to compare species abundance at multiple sampling sites across a heterogeneous landscape with various environmental factors [48]. Several methods are used for this analysis, including latent factor mixed models (LFMM; [49]), redundancy analysis (RDA; [50]), and gradient forest [51]. Gradient forest may be the most advanced method, as it employs multiple regression trees to estimate environmental thresholds that drive community turnover across the landscape. This method, which extends the principles of random forest to handle multiple response variables, allows for the detection of both linear and nonlinear relationships between communities and their environments, while also controlling for collinearity among environmental variables [51]. In our study, we apply the gradient forest method to identify the environmental predictors of coral community composition across the reefs of St. Croix in the U.S. Virgin Islands.

Abundances of each lineage within each species at each site were used to produce a community-by-site table for investigating the influence of environmental gradients on cryptic lineage composition. These counts became our response matrix that was input to a gradient forest model with the environmental predictors (using R package *gradientForest* version 0.1.32, [51]). We ran two gradient forest models: one with depth and assignment to four ecoregions and one with all 40 environmental predictors. Random forests were grown for each lineage, each with an ensemble of 500 trees, where each tree splits environmental gradients at different observations. The change in community composition across each split was then summarized into the compositional turnover along each environmental gradient. The importance of each predictor was computed with cross-validation and assessed by conditional permutation of each variable, permuted with a maximum of two splits on predictors correlated > 0.5.

## Results

### Cryptic genetic lineages

After quality filtering individual corals and genomic loci, a subset was retained for analyzing population structure within each species (Table 1, S4 Table in S1 File). Technical replicates and genetically identical individuals were removed from the final subset by retaining the sample with highest alignment rate to its respective coral genome.

We identified two genetic lineages within *A. agaricites*, *O. faveolata*, and *P. astreoides* (Fig 3A,3C and 3D) and four lineages of *P. strigosa* based on visual examination of the hierarchical clustering tree and PCoA of the identity-by-state genetic dissimilarity matrix for each species. When clustering *A. agaricites* and *P. astreoides* with samples from a prior study in Florida [43], we observed that the two *A. agaricites* lineages from St. Croix cluster with two genetic lineages found throughout the Florida Keys. Specifically, the red St. Croix lineage (Fig 3A) clusters with

**Table 1. Sample size and genomic sites retained for six coral species.**

| Species | # individuals retained after quality filtering and removing clones | # geographic sites represented | # genomic sites retained after sequencing |
|---|---|---|---|
| *A. agaricites* | 25 | 8 | 49,992 |
| *M. cavernosa* | 35 | 8 | 32,790 |
| *O. faveolata* | 47 | 10 | 28,724 |
| *P. astreoides* | 47 | 6 | 115,034 |
| *P. strigosa* | 70 | 11 | 35,943 |
| *S. siderea* | 21 | 8 | 49,863 |

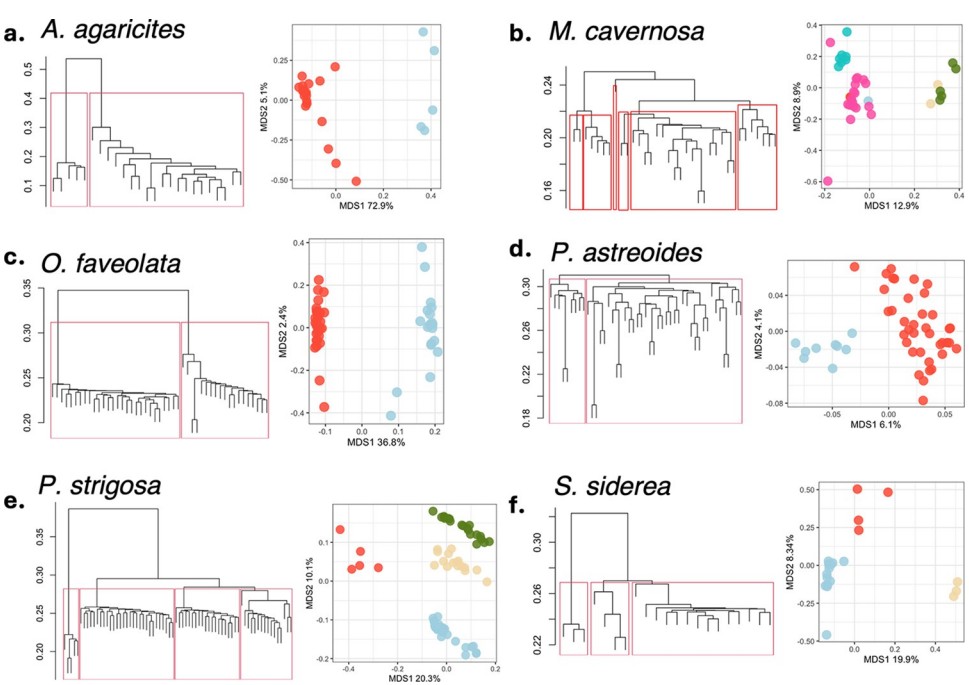

**Fig 3. Identifying cryptic genetic lineages in six coral species in St. Croix.** (a) *A. agaricites* is composed of two distinct genetic lineages as shown in the hierarchical clustering tree (left) and PCoA (right) of genetic dissimilarities. (b) *M. cavernosa* contains six lineages, as confirmed by clustering with a larger sample set from Florida and the Gulf of Mexico. (c) *O. faveolata* contains two lineages, (d) *P. astreoides* contains two lineages, (e) *P. strigosa* contains four lineages, and (f) *S. siderea* contains three lineages.

the shallow-preferred lineage in Florida and the blue St. Croix lineage clusters with the depth-generalist lineage in Florida. However, the two *P. astreoides* lineages from St. Croix do not cluster with any of the lineages observed in the Florida Keys, and instead appear to be genetically differentiated. The *M. cavernosa* and *S. siderea* sample sets also clustered with samples from prior studies [21, 42, 52]. *M. cavernosa* from St. Croix clustered with six lineages specialized to various depths, and *S. siderea* clustered with two shallow and one deep-specialized lineage found across the Florida Keys and the Gulf of Mexico. Differentiation of three *S. siderea* lineages is apparent in hierarchical clustering and PCoA (Fig 3F), but the delineation of six *M. cavernosa* lineages is less obvious, especially in ordination space (Fig 3B). However, this is likely due to the under-sampling of *M. cavernosa* around St. Croix, and a larger sample set would likely reveal more striking differentiation of lineages.

Cryptic lineages within each coral species show different geographic distributions. For instance, the red lineage of *A. agaricites* occurs at all sites, but the blue lineage was only found at one western site (Fig 4A). Similarly, the red lineage of *P. astreoides* occurs at all sites but the blue lineage was only found at one central site (Fig 4D). On the other hand, the two lineages of *O. faveolata* seem to be geographically segregated and do not co-occur at the same sites (Fig 4C). *P. strigosa*, which contains four lineages, shows geographic partitioning of the green lineage to the west and blue lineage in the east (Fig 4E). Similarly, *S. siderea* shows partitioning of the red and tan lineages in the central sites and outer sites, respectively. However, the blue lineage *S. siderea* occurs at all sites (Fig 4F). *M. cavernosa*, which contains six lineages, also shows some geographic separation, as the red, green, and tan lineages only occur at central sites while the pink and turquoise lineages occur everywhere (Fig 4B).

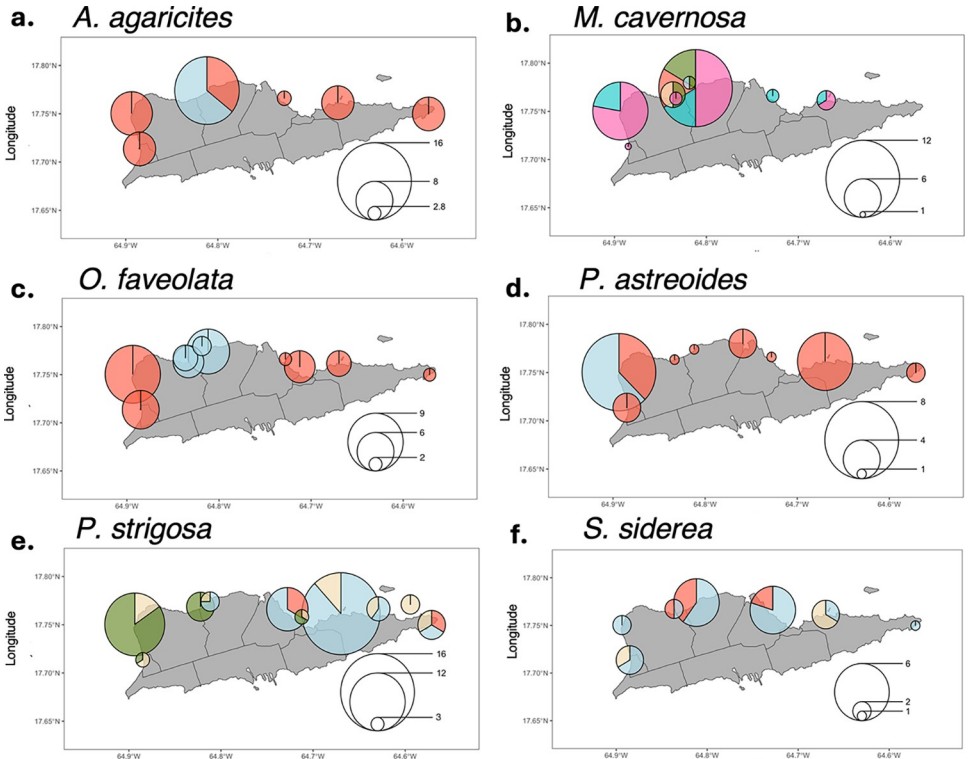

**Fig 4. Spatial distribution of cryptic genetic lineages in six coral species from St. Croix.** Each pie chart represents the occurrence of distinct genetic lineages at each sampling site, and the size legend in the bottom right of each panel indicates the number of samples in each pie. Different lineages are indicated by color and correspond to points in PCoAs from Fig 2. Coastlines and district boundaries are plotted with GADM mapping data (https://gadm.org/index.html).

## Community ecology

In the first gradient forest model, depth and ecoregions together accounted for 15.4% of the variation in cryptic coral communities. Depth was the strongest predictor (cross-validation $R^2$ = 0.083) and Ecoregion B, representing central St. Croix near the capital Christiansted (Fig 2C), was the most important ecoregion driving community structure (Fig 5A). When summing importances of all ecoregions together (cross-validation $R^2$ = 0.071), they explain less community structure than depth. These findings imply that depth and ecoregions both contribute to the structure of cryptic coral communities, but depth is a more important predictor than ecoregions.

When evaluating the importances of all 40 environmental variables, they together accounted for 29.5% of the community composition. Depth was the strongest predictor (Fig 5B, $R^2$ = 0.15), and the yearly range of pH was the second most important (Fig 5B, $R^2$ = 0.011). Depth shows a prominent ecological threshold around 5 meters deep, where there is a sharp turnover of community assembly (Fig 5B). Notably, depth increases with distance from shore at all sites (Fig 5E), though eastern sites associated with Ecoregion A are generally the shallowest (Fig 2E). We also observed a gradual turnover when the yearly range of pH is 0.4–0.5 and steep turnover around 0.5 (Fig 5C), which corresponds to Ecoregion B on the map (Figs 2C and 5F).

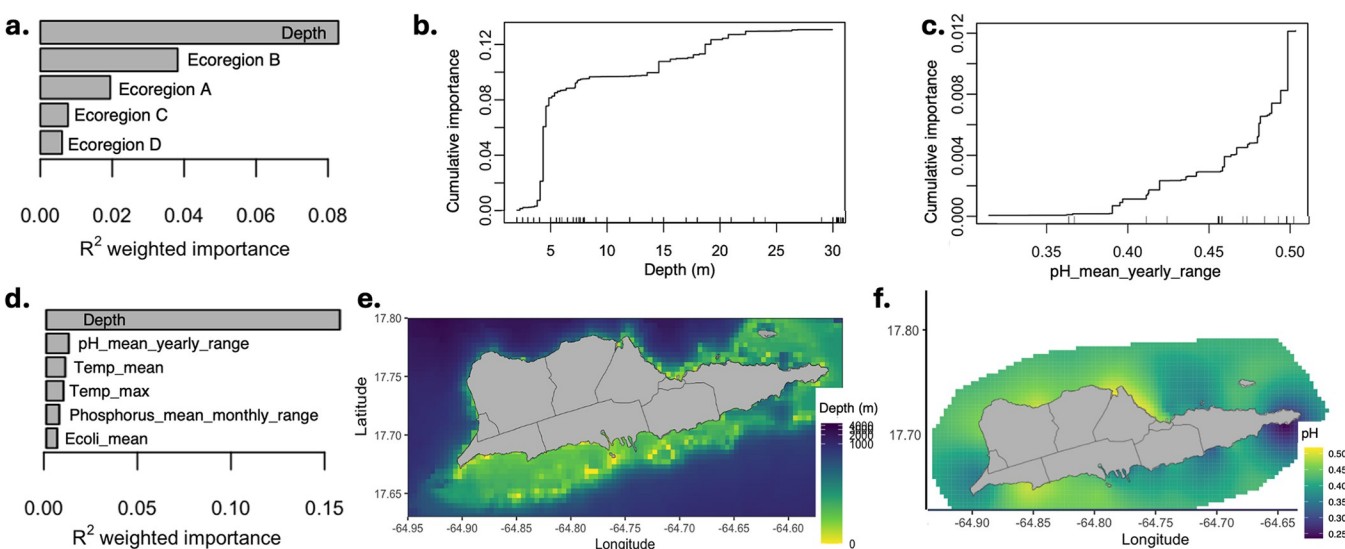

**Fig 5. Environmental drivers of cryptic coral communities.** (a) Estimated importance (cross-validation $R^2$) of depth and four ecoregions to coral community composition across the seascape. (b) Top five out of 40 environmental predictors, based on cross-validation $R^2$ in a gradient forest model. (c-d) Cumulative importance distributions show change in community composition along the range of the two most important variables. Tick marks on the x-axis indicate the environmental values at each site. (c) Coral communities demonstrate steep turnover around 5 meters deep, and (d) communities show graduate turnover across a gradient of yearly pH range. (d) Map of depths around St. Croix, using bathymetry data from GEBCO 2023 Grid (doi:10.5285/f98b053b-0cbc-6c23-e053-6c86abc0af7b). (e) Map of the maximum temperature (averaged from 2010–2022) and derived from interpolation of *in situ* monitoring data from the Virgin Islands DPNR. Coastlines and district boundaries are plotted with GADM mapping data (https://gadm.org/index.html).

## Discussion

Multiple cryptic genetic lineages were detected within six coral species from St. Croix, revealing a complex genetic landscape. These include at least two lineages within *A. agaricites*, *P. astreoides*, and *O. faveolata*, three within *S. siderea*, four within *P. strigosa*, and six within *M. cavernosa*. The geographic partitioning observed among some of these lineages suggests that local reef environments may influence cryptic genetic variation. When assessing the abundance of these 19 cryptic lineages, it becomes evident that community composition is partially determined by both depth and the natural environmental boundaries that define distinct ecoregions. However, we note that these associations reflect how community structure correlates with long-term environmental trends- in this case, abiotic variables summarized over 9–22 years. Further investigations into the role of instantaneous selective pressures, such as cyclones or bleaching events, could be conducted by resampling coral over time. Comparing abundance before and after such events could provide insights into how cryptic communities shift in response to these acute pressures, thereby enhancing our temporal understanding of coral responses to changing environmental conditions.

The cryptic variation identified in this study is consistent with previous findings in most of the species examined. Depth-partitioned cryptic lineages have been reported in *Agaricia* [19] and specifically in *A. agaricites* [43], as well as in *P. astreoides* [43, 44, 53], *S. siderea*, and *M. cavernosa* [21, 42, 52]. These studies consistently demonstrate that genetic lineages exhibit preferences for specific depths rather than strict depth boundaries. Depth partitioning was also identified in *O. faveolata* lineages from Puerto Rico [54], although previous research from Panama detected three lineages with variations in thermal tolerance [16]. *P. strigosa* remains relatively understudied in population genetics, despite its high morphological variability suggesting potential for cryptic genetic variation [55]. The four *P. strigosa* lineages identified in this study represent the first documented instances of cryptic genetic divergence within this

species. Overall, the coral reefs of St. Croix exhibit genetic diversity comparable to other regions in the Caribbean.

Our investigation into the associations between community composition and environmental factors revealed that ecoregions may play a supporting role in shaping cryptic community structure, second to depth. Previous studies have also identified depth as a primary factor influencing cryptic variation [17, 19, 21, 56]. However, our findings underscore the novel impact of ecoregions on the structuring of cryptic lineages. When re-analyzing community-environment associations using all abiotic variables employed to define ecoregions, depth re-emerged as the most important predictor, followed by various measures of pH, temperature, and other variables (Fig 5D). While depth and temperature are plausible candidates for driving community structure, the actual variables that drive ecoregion differences, beyond depth, may not be represented in our data. Our findings suggest that unique local conditions within each ecoregion collectively shape the structure of cryptic coral communities, such that nuanced shifts in many environmental factors drive community divergence.

While our incorporation of cryptic genetic lineages into a multi–species study represents a novel application to coral community ecology, previous research has identified environmental drivers of coral communities at the species level. Important drivers include cyclones [57], temperature anomalies [57–59], productivity [60], latitude [61], sedimentation [62], and depth [63]. Although depth emerges as a common driver across many coral communities worldwide, a study in the Indo-Pacific region found it to be only a minor predictor for two coral species [59]. One potential explanation for the prevalence of depth as a driver could be its impact on light reduction [64] and algal biomass and diversity [65]. For instance, a study of coral reefs in South Africa delineated depth thresholds at 15 meters for *Pocillopora damicornis* and 33 meters for reef communities [61], mirroring the depth threshold we observed at approximately 14–22 meters (see Fig 5C). Additionally, previous research in the Virgin Islands noted variations in community structure among different sites [66], which aligns with our finding that ecoregions may exert influence over community composition.

One factor (beyond depth) that might have a direct effect on the cryptic coral community is the yearly range of pH, the next most important predictor after depth (Fig 5D). It shows notable differences within Ecoregion B (see Fig 5E). After pH, mean and maximum temperature appear to be the next most drivers of cryptic coral community structure (Fig 5B). In St. Croix, maximum temperature shows notable differences near Ecoregion B and Buck Island National Monument (S3h Fig in S1 File). Fluctuations in pH [67] and temperature [68–70] have been shown to affect coral reef ecosystem structure and function, so their putative role in driving community structure on St. Croix seems likely. However, while we can speculate about how pH, temperature, and other variables may impose selective pressures, it is essential to consider the multicollinearity present in our environmental dataset (Fig 2A). Therefore, although our findings align with our hypotheses that depth and temperature are critical environmental drivers influencing genetic divergence in cryptic communities, we view these variables as part of the unique local conditions within each ecoregion that collectively shape selection pressures on cryptic coral communities.

While we can estimate the extent of environmental differences that contribute to community structure, we recognize that most lineages are not strictly confined to specific depths or ecoregions and can co-occur at certain sites (see Fig 4). Untangling the mechanisms driving and maintaining genetic differentiation between cryptic lineages in the absence of geographic isolation remains a challenge. In addition to environmental variables, non-environmental factors such as ocean currents, reproductive strategies, natural disturbances, and prezygotic barriers may also influence coral community structure. For example, ocean currents can restrict dispersal patterns of local coral taxa [71] and determine the settling locations of coral larvae

[72]. In St. Croix, currents bend around the southern coast, sending wake flows to both the eastern and western sites [45], which may help explain the structured distribution of cryptic lineages across the east and west. Additionally, the reproductive mode of coral species- whether brooding or spawning- influences dispersal distance due to variations in larval phase length. For instance, the broadcast spawner *Acropora palmata* can have parent-offspring separations of 70 meters to one kilometer [73], while brooding *Agaricia* corals typically only disperse 2 to 11 meters per generation [74]. Despite the influence of ocean currents, many coral larvae settle in close proximity to their parent colonies [74], suggesting that short dispersal distances may limit the role of currents in driving genetic differentiation within these populations.

Natural disturbances, such as hurricanes, can also shape the structure of cryptic coral communities by fragmenting corals and redispersing them over large distances. However, the impact of these disturbances varies depending on oceanic geographic features. For example, patterns of reef destruction in St. Croix caused by Hurricane Hugo in 1989 varied by depth, shoreline orientation, and the composition of benthic communities before the storm [75]. Moreover, patterns of coral growth on the mesophotic shelf edge of the U.S. Virgin Islands appear to be structured by acute but infrequent swell impacts, which varies across depths [76]. These examples illustrate how natural disturbances can generate distinct patterns in coral community structure, shaped by the interaction between damage and recovery processes. On St. Croix, the impacts of Hurricanes Irma and Maria were similarly devastating, with significant damage to heritage and fisheries resources [77, 78], though the full scope of damage to the island's coral communities remains unclear.

Prezygotic barriers, such as temporal isolation and gamete incompatibility, can also drive genetic divergence and the formation of cryptic lineages in corals. Within a single species, individuals may spawn at different times of the year, with some spawning in spring and others in fall. This asynchronous spawning is observed in species such as *Acropora tenuis*, *A. samoensis*, *A. digitifera*, *Orbicella spp.*, and *Mycedium elephantotus*, all of which show genetic differentiation linked to their spawning periods [79–83]. Spawn timing is also determined by environmental cues and genetics, and even just a few hours difference in spawning can lead to sympatric speciation [84]. Gamete incompatibility further contributes to this process. In some *Orbicella spp.*, eggs can demonstrate conspecific sperm precedence (CSP), where they preferentially accept sperm from their own species. CSP may help gametes from broadcast spawners find each other in the water column, but it may also drive divergence in gamete compatibility [85]. In the Caribbean, three recently diverged *Orbicella spp.*- *O. franksi*, *O. faveolata*, and *O. annularis*- show varying levels of gamete incompatibility, which may have played a role in their speciation [86].

Despite these alternative explanations for the observed distribution patterns, depth consistently appears to be a significant driver of cryptic genetic differentiation in prior literature [20] and in our study. In St. Croix, depth can explain geographically proximate yet genetically divergent coral populations. This is particularly evident in *O. faveolata*, *S. siderea*, *M. cavernosa*, and *P. astreoides*, all of which exhibit high lineage diversity along the extensive depth gradient in Ecoregion C (see Figs 2 and 4). However, further exploration is warranted to investigate potential hybrid zones between cryptic lineages present on both sides of this ecological barrier, and to examine non-environmental drivers such as those described above.

Although cryptic communities in St. Croix are associated with depth and ecoregions, it is uncertain whether these factors have a causal relationship with community assemblage. However, these associations with significant predictors can be empirically tested through field experiments. Reciprocal transplantations across ecoregion boundaries offer a means to examine local adaptations by assessing the fitness (i.e., survival or growth) of transplanted lineages.

For instance, a reciprocal transplantation experiment involving five coral species from shallow (5-10m) and deep (45m) sites demonstrated decreased fitness of corals from deep sites when transplanted to shallow sites [87]. Similar experiments within our study system could ascertain whether cryptic lineages can thrive when outplanted beyond their native depth or ecoregion boundaries to aid in reef restoration initiatives. In addition, common garden experiments conducted *ex situ* could validate the influence of identified environmental predictors, such as temperature thresholds, on fitness. For instance, a common garden experiment investigating *Acropora pulchra* found that elevated temperatures and increased $pCO_2$ levels led to reduced growth, suggesting these variables likely shape the distribution of this species across the seascape [88].

Continued efforts to characterize cryptic variation within coral species and understand the unique environments supporting genetically distinct populations will be crucial for informing effective coral outplanting strategies [20]. As restoration programs begin to identify resilient genotypes for propagation, insights from cryptic lineages and their evolutionary trajectories can guide the spatial planning of coral outplants, ensuring they are placed in environmental conditions optimal for their survival [13].

## Conclusions

In this study, we observed that cryptic genetic lineages within many coral species form distinct communities that vary across depth and ecoregions. Given that all six species and at least 11 cryptic lineages in this study are distributed across the Caribbean, these findings could be generalizable beyond the Virgin Islands. As human impacts on coral reefs escalate, evaluating the direct effects of environmental changes on coral fitness, particularly in the Caribbean, becomes increasingly crucial. Future studies directly assessing the impact of the identified predictors will be essential in determining the adaptability or restriction of cryptic lineages to specific conditions. Furthermore, characterizing environmental heterogeneity across the seascape and understanding its influence on cryptic communities will be vital for guiding future restoration efforts.

## Supporting information

**S1 File.**
(ZIP)

## Acknowledgments

We like to thank everyone at The Nature Conservancy's Coral Innovation Hub in St. Croix, USVI for their guidance and camaraderie in the field. We would especially like to thank Ashlee Lillis, Emily Klosterman, Emily Nixon, Robin Smith, Moose Marusa, Matthew Warham (DPNR), Carly Scott (UT-Austin) and Daisy Flores (UT-Austin) for helping with fieldwork for this study. We would also like to thank Cruzan Rum for providing ethanol for laboratory work on island. The bioinformatic analyses were performed using the high-performance computing resources of the Texas Advanced Computing Center (TACC).

## Author Contributions

**Conceptualization:** Mikhail V. Matz, Kristina L. Black.

**Data curation:** Gia N. Cabacungan, Tharani N. Waduwara Kankanamalage, Amilah F. Azam, Madeleine R. Collins, Abigail R. Arratia, Kristina L. Black.

**Formal analysis:** Gia N. Cabacungan, Tharani N. Waduwara Kankanamalage, Amilah F. Azam, Madeleine R. Collins, Abigail R. Arratia, Kristina L. Black.

**Funding acquisition:** Kristina L. Black.

**Investigation:** Gia N. Cabacungan, Tharani N. Waduwara Kankanamalage, Amilah F. Azam, Madeleine R. Collins, Abigail R. Arratia, Kristina L. Black.

**Project administration:** Alexandra N. Gutting, Mikhail V. Matz, Kristina L. Black.

**Supervision:** Alexandra N. Gutting, Mikhail V. Matz, Kristina L. Black.

**Visualization:** Gia N. Cabacungan, Tharani N. Waduwara Kankanamalage, Amilah F. Azam, Madeleine R. Collins, Abigail R. Arratia, Kristina L. Black.

**Writing – original draft:** Gia N. Cabacungan, Tharani N. Waduwara Kankanamalage, Amilah F. Azam, Madeleine R. Collins, Kristina L. Black.

**Writing – review & editing:** Gia N. Cabacungan, Tharani N. Waduwara Kankanamalage, Amilah F. Azam, Madeleine R. Collins, Abigail R. Arratia, Alexandra N. Gutting, Mikhail V. Matz, Kristina L. Black.

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
