## [Decision Letter · Decision Letter 0]

17 Sep 2024

PONE-D-24-30961Cryptic coral community composition across environmental gradientsPLOS ONE

Dear Dr. Black,

Thank you for submitting your manuscript to PLOS ONE. After careful consideration, we feel that it has merit but does not fully meet PLOS ONE’s publication criteria as it currently stands. Therefore, we invite you to submit a revised version of the manuscript that addresses the points raised during the review process.

We look forward to receiving your revised manuscript.

Kind regards,

Vitor Hugo Rodrigues Paiva, Ph.D.

Academic Editor

PLOS ONE

Journal Requirements:

2. Thank you for stating the following financial disclosure: This study was funded by The Nature Net Fellowship from The Nature Conservancy to K.L.B..

3. We note that Figures 1 and 2 in your submission contain [map/satellite] images which may be copyrighted. All PLOS content is published under the Creative Commons Attribution License (CC BY 4.0), which means that the manuscript, images, and Supporting Information files will be freely available online, and any third party is permitted to access, download, copy, distribute, and use these materials in any way, even commercially, with proper attribution. For these reasons, we cannot publish previously copyrighted maps or satellite images created using proprietary data, such as Google software (Google Maps, Street View, and Earth). For more information, see our copyright guidelines: http://journals.plos.org/plosone/s/licenses-and-copyright.

a. You may seek permission from the original copyright holder of Figures 1 and 2 to publish the content specifically under the CC BY 4.0 license.  

Reviewers' comments:

Reviewer's Responses to Questions

**Comments to the Author**

1. Is the manuscript technically sound, and do the data support the conclusions?

Reviewer #1: Yes

Reviewer #2: Yes

2. Has the statistical analysis been performed appropriately and rigorously? 

Reviewer #1: Yes

Reviewer #2: Yes

3. Have the authors made all data underlying the findings in their manuscript fully available?

Reviewer #1: Yes

Reviewer #2: Yes

4. Is the manuscript presented in an intelligible fashion and written in standard English?

Reviewer #1: Yes

Reviewer #2: Yes

5. Review Comments to the Author

Reviewer #1: General comments

Evaluating how population and community structure is influenced by environmental conditions is key to understanding the influence of environmental change on ecosystems and implementing effective restoration/conservation efforts. This study explores the genetic structure of coral communities around St. Croix and how that genetic structure relates to a spatiotemporal variation in environmental conditions. The authors find variable genetic structuring across coral taxa with certain environmental conditions of key influence on coral community structure over space and time.

Generally, I found this manuscript to be well-written, well-structured and easy to read. Moreover, the quantitative design appears to be well constructed and performed. However, I feel that the manuscript would benefit from greater detail on portions of the analysis and environmental data structure (expanded on below). My technical understanding of these genetic methods is limited and thus I cannot provide meaningful technical review of this section.

Although the study is designed to be a more exploratory assessment, an important consideration here is the lack of specific hypotheses that are set out to be tested. While this isn’t necessarily a make-or-break issue, it is important for how the study is framed. The authors suggest that the broader implication of this work is around conservation and restoration actions—understanding where corals perform well and poorly can inform where corals are out planted. However, I find the interpretation of results provided here to be a bit of a stretch in a restoration context. Simply put, it is difficult to assess whether corals perform better or worse in a particular area without causal-type experimentation and hypothesis testing, indeed the authors indicate that such examination is a logical next step. This association-type analysis evaluates how spatial variation in genetic structure of corals is associated with long-term trends in environmental conditions, but it doesn’t say how these variables actually affect corals or their genetic structure. Given that consideration, reframing portions of the introduction and discussion to be focused on how environmental conditions can contribute to genetic heterogeneity might be a better framing than trying to make the leap regarding restoration efficacy.

Specific comments

Introduction

L53-55: “ecologically cohesive communities”. What is an ecologically cohesive community? Does this imply some type of facilitation by heterospecifics? This is not defined here or in the referenced literature. The potential facilitation effect may not be entirely relevant to this study.

L60: The authors note species and genera. Better to note these as coral “taxa” not species.

L64: The referenced literature, Grupstra et al. (2024 in NEE), reviews the potential drivers of genetic divergence among cryptic corals. I wouldn’t necessarily call this divergence for unknown reasons as the authors purport in this line. Rather it appears that there are many supported hypotheses for why provided by Grupstra et al. and by the authors in the following sentence. This context is a critical component of this study given that the authors do not generate specific hypotheses regarding why certain environmental attributes influence genetic structure. Perhaps a more useful approach would be to evaluate the hypotheses, pre or post-hoc, generated by Grupstra et al. within the study system, species.

L77-79: I appreciate the broadening of this topic to relevant examples in temperate ecosystems. However, I would suggest focusing this section on the abiotic factors that influence marine communities exclusively. Perhaps the broader strokes would be more useful earlier in the introduction.

L89-91: “highlighting the intricate relationship between genetic variation and environmental challenges in coral ecosystems.” Can the authors be more specific regarding environmental challenges? Are they referring to climate change?

L93-104: This paragraph seems like it should be in the methods section. The authors note potential mechanistic drivers of these patterns in the previous paragraph. Perhaps this space could be used to introduce specific hypotheses that the authors are testing?

Methods

L153-155: This passage would benefit from additional detail regarding sampling design. What was the size distribution per species of colonies sampled? Are these similarly sized colonies? What is the sample size per species per site?

L222-223: Can the authors expand on the structure and processing of environmental data? Mainly, what was the temporal resolution of environmental data used and subsequently summarized? Perhaps this information can be added to Supplementary Table 1. Moreover, interpretation of the environmental heterogeneity across sites and years would benefit from an additional supplementary figure illustrating spatiotemporal variation in each environmental variable.

L227: Kriging interpolation appears to be an appropriate way to interpolate environmental conditions for the location of colonies sampled. However, evaluating interpolation performance is contingent on the location of monitoring sites relative to colony locations. Where were the environmental monitoring sites? How far apart were they? Can the authors provide a supplementary figure of the variograms generated from kriging interpolation?

L235-236: The authors note that the “optimal number of genetic linkages was visually determined” (L206). Was a similar method performed for the environmental data (i.e., ecoregions and environmental data)? For ecoregions, what was the visual criteria for ecoregion grouping? Was there consideration of the height of each site similarity?

L260: It appears that two gradient forest models were employed here: one that assessed the influence of the 40 environmental predictors and another that assessed the effect of depth and the four ecoregions. Is this interpretation correct? This section would benefit for an explicit statement regarding the model structure[s].

Results

Table 1: What were the sample sizes per species per location? Are the sample sizes adequate to characterize genetic variation within and between locations? Additionally, in Fig. 4, there is some indication that genetic diversity appears to increase with sample size (panel b., M. cavernosa) although these patterns may not be as evident for the other 5 taxa. It is possible that the diversity of genetic linkages revealed may be an artifact of sampling design, although it is challenging to assess this without explicit reporting of sampling size per site per species. Do sample sizes reflect the relative abundance of each taxa at each site?

L338: The authors refer to Fig. 1d here. Should this be Fig. 2c? Its difficult to tell.

L334: The amount of variation in coral community structure explained by environmental conditions in the two separate models appears to be relatively small (12.5 and 18.9%), with cross validation values quite low for environmental predictors? Could this be a result of overfit models (i.e., too many environmental predictors)? Can the authors provide detail as to whether hierarchal cluster models and subsequently gradient forest models were assessed with fewer environmental variables? A more parsimonious environmental cluster model could contribute to increased gradient forest model performance. It might be beneficial to provide an explanation of why the variance explained was relatively low (e.g., low sample size).

L346: The authors refer to depth being the strongest predictor of community composition. However, Fig. 5a shows Ecoregion to be the most important, followed by depth. Do they mean Fig. 5b?

Figure 4: Add figure legend for bubble size corresponding to sample size.

Figure 5: Remove titles from panels A and B and provide x-axis labels.

Discussion

L376: The authors note that community composition is partially shaped by environmental conditions, mainly ecoregions and depth. Indeed, there is a high degree of variation in coral community structure that isn’t explained by models. While the authors expand on other potential drivers of these dynamics (e.g., cyclones temperature anomalies etc.), it is important to note that the temporal resolution of environmental conditions is not matched to the instantaneous measure of genetic structure. In other words, the authors assess how the environmental legacy of sites over ~12 years contributes to “current” community genetic structure. This approach makes many assumptions about what the coral community ‘looked like’ over this 12-year period. An acknowledgement and brief discussion of this assumption is probably needed here.

Reviewer #2: This is a nice study and a well-written paper. Strengths are that the distribution of cryptic coral lineages were studied in many (six) sympatric species from many (12) sites. One potential weakness might be that the number of individuals sampled per species was perhaps a bit low in some species (n = 21, 25 spread across 8 sites) so potential that some composition-environment patterns might not be fully characterized (but this is hard to know).

The manuscript is presented well, and it was easy to read. The data analyses all seem robust to me. I think the result will be useful to those working at St. Croix as well as to the bolster the evidence that cryptic species of corals are becoming the rule rather than the exception.

The overall result is consistent with previous studies that many morphologically-defined species contain evolutionarily distinct genetic lineages, that these lineages commonly co-occur but differ in relative abundance across depths and locations that differ in environmental variables. This study finds a correlation with yearly pH variation.

My only main comments are:

1) Because the association between community composition and environmental variables is still a correlation (and this is acknowledged e.g., 459), the discussion could perhaps be strengthened by considering alternative explanations for the current composition-distribution pattern that can be made without invoking environmental variables, like dispersal patterns / limitation or that they are just a signature of past disturbance history.

2) Genetic lineages within species were identified by visual examination only (of the hierarchical clustering tree and PCoA). The clustering analyses included samples from prior studies, which increases the confidence in the visual determination of lineages. However, the evidence for how the current samples cluster with previous samples is hard to find. It might be good to show which tips in Figure 3 are the “reference” samples from previous studies, and make a better connection to the nomenclature of each lineage (if a consistent one even exists for these species?), so readers can trace what lineages are showing up where.

Minor:

Line 224: Could you mention here what the frequency of recording are, and what the mean, maximum, etc refer to (mean per day, per month? How many measurements comprised each mean?). This is just to get a better idea of how well the environmental variables are characterized.

Line 227: Again, could you mention the spatial resolution of the sampling (how many sites?). Just trying to understand the robustness of the spatial interpolations. If a site where coral is sampled is far from the nearest site where environmental variables were measured, how good are the environmental estimates?

6. PLOS authors have the option to publish the peer review history of their article (what does this mean?). If published, this will include your full peer review and any attached files.

Reviewer #1: No

Reviewer #2: No

---

## [Author Response · Author response to Decision Letter 0]

19 Nov 2024

Thank you for the opportunity to revise our manuscript. In response to the reviewer’s concerns, we’ve made the following important changes:

1. We reframed our manuscript so that it now presents a hypothesis in the introduction that is grounded in prior literature. Then we revisit our hypothesis in the discussion, to show how our results compared to prior research.

2. We added many more tables and figures to the supplementary material, per both the reviewer's requests. The addition of this information greatly improves our ability to illustrate the environmental heterogeneity around St. Croix.

3. In producing the new supplementary tables and figures, we found some spurious environmental observations that might have skewed our interpolations. So we cleaned up the environmental input by removing months or years with only singular observations, re-interpolated the variables, and then re-conducted the gradient forest model analyses. The results were mostly the same with only two differences- the sample collection site Columbus Landing is now in Ecoregion B (which actually makes more logical sense given its location), and the second most important environmental predictor of coral communities became maximum temperature. We changed the main text figures as well, to reflect this.

We believe these changes greatly strengthen our manuscript and hope that the editor and reviewers will agree. Please see our reviewer response document to see our full responses to all reviewer's concerns line-by-line.

Thank you on behalf of all co-authors,

Kristina Black

---

## [Decision Letter · Decision Letter 1]

5 Dec 2024

PONE-D-24-30961R1Cryptic coral community composition across environmental gradientsPLOS ONE

Dear Dr. Black,

Thank you for submitting your manuscript to PLOS ONE. After careful consideration, we feel that it has merit but does not fully meet PLOS ONE’s publication criteria as it currently stands. Therefore, we invite you to submit a revised version of the manuscript that addresses the points raised during the review process.

We look forward to receiving your revised manuscript.

Kind regards,

Vitor Hugo Rodrigues Paiva, Ph.D.

Academic Editor

PLOS ONE

Journal Requirements:

Reviewers' comments:

Reviewer's Responses to Questions

**Comments to the Author**

1. If the authors have adequately addressed your comments raised in a previous round of review and you feel that this manuscript is now acceptable for publication, you may indicate that here to bypass the “Comments to the Author” section, enter your conflict of interest statement in the “Confidential to Editor” section, and submit your "Accept" recommendation.

Reviewer #1: (No Response)

Reviewer #2: (No Response)

2. Is the manuscript technically sound, and do the data support the conclusions?

Reviewer #1: Yes

Reviewer #2: Yes

3. Has the statistical analysis been performed appropriately and rigorously? 

Reviewer #1: Yes

Reviewer #2: Yes

4. Have the authors made all data underlying the findings in their manuscript fully available?

Reviewer #1: Yes

Reviewer #2: Yes

5. Is the manuscript presented in an intelligible fashion and written in standard English?

Reviewer #1: Yes

Reviewer #2: Yes

6. Review Comments to the Author

Reviewer #1: Evaluating how population and community structure is influenced by environmental conditions is key to understanding the influence of environmental change on ecosystems and implementing effective restoration/conservation efforts. This study explores the genetic structure of coral communities around St. Croix and how that genetic structure relates to a spatiotemporal variation in environmental conditions. The authors find variable genetic structuring across coral taxa with certain environmental conditions of key influence on coral community structure over space and time.

This is my second review of this manuscript. The authors have made useful modifications and additions to all sections of the manuscript that assist in bolstering their approach. The revised version of this manuscript is much improved. Although the hypothesis outlined in the introduction is relatively broad, I feel that the updated context of the approach assists in the interpretation of the results. I have a few lingering comments that should be addressed and may benefit the manuscript further.

L89-100: In the previous version of the manuscript I noted that this paragraph would be best if placed in the methods section. Although the revised introduction adequately addresses my other comments I still feel that a description if the specific gradient forest methods here is out of place. Although these methods are certainly key to the analytical structure of the assessment, noting them in the introduction feels abrupt, especially after the paragraph noting the hypothesis examined but before the description of study area. Given that the gradient forest methods aren’t the key focus of this study, rather environmental drivers of cryptic coral composition, the introduction would benefit from a reduced focus on the particular methods.

Supplementary Figures: These appear to be out of order in the main text. Although this will probably be addressed in copy-editing, it may be best to rearrange them at this stage.

Semivariograms and kriging approach:

I appreciate the authors providing these supplementary figures—they greatly enhance the interpretation of the spatial interpolation of environmental conditions. Although, it is unclear what the numbers next to each point represent. Also, shouldn’t the points be pairwise comparisons of environmental stations? If so, shouldn’t there be many more points for each plot which may improve model fit.

As the authors note, the fit of these models is quite variable: some perform better than others. Based on even the good fits, I am somewhat concerned about the spatial scale of interpolation. For example, the fit for DO_max isn’t very good over the distance of ~0.03 (what are the units here?). For other variables (e.g., monthly range in Nitrogen), the fit line doesn’t appear to fit the [negative] trend at all. Given these fits, I am concerned that the environmental patterns over space may not be adequately represented by the models described. In general, some of the fits appear to either (1) obscure the spatial variability inherent to some variables or (2) don't appropriately capture trends.

Reviewer #2: 1) Overall I thought the revised text was fairly superficial, but it more or less addresses the point. For example, the phrase “stepped clines driving tension zone formation and assortative mating” is quite jargony and “other factors, such as ocean currents influencing larval dispersal” is quite vague. The statement that “the ability of currents to sustain high dispersal and connectivity” (L462) seems at odds with all the research showing that larvae (including corals) disperse much less than expected from ocean currents. For example: Prata et al 2024 Some reef-building corals only disperse metres per generation. Proc. R. Soc. B 291: 20231988. https://doi.org/10.1098/rspb.2023.1988

2) Thanks for the info. I noted that no changes were made to the manuscript here. The point of my comment was just to make some edits that could help the reader (not just the reviewer). I think at least a statement in the manuscript that summarizes these connections would help the reader.

“Generally, environmental variables were monitored across all coastlines of St. Croix, and very close to coral sampling sites. The only exception is that E. coli was not monitored near Cane Bay on the north shore.” So this would be good to mention this is the manuscript.

7. PLOS authors have the option to publish the peer review history of their article (what does this mean?). If published, this will include your full peer review and any attached files.

Reviewer #1: No

Reviewer #2: No

---

## [Author Response · Author response to Decision Letter 1]

17 Jan 2025

Note: Please see separately attached document containing these responses to reviewer comments (formatted for easier reading). This document is also included at the end of the compiled PDF that contains all manuscript components.

Reviewer Comments to Author 

Reviewer #1: 

Evaluating how population and community structure is influenced by environmental conditions is key to understanding the influence of environmental change on ecosystems and implementing effective restoration/conservation efforts. This study explores the genetic structure of coral communities around St. Croix and how that genetic structure relates to a spatiotemporal variation in environmental conditions. The authors find variable genetic structuring across coral taxa with certain environmental conditions of key influence on coral community structure over space and time.

This is my second review of this manuscript. The authors have made useful modifications and additions to all sections of the manuscript that assist in bolstering their approach. The revised version of this manuscript is much improved. Although the hypothesis outlined in the introduction is relatively broad, I feel that the updated context of the approach assists in the interpretation of the results. I have a few lingering comments that should be addressed and may benefit the manuscript further.

L89-100: In the previous version of the manuscript I noted that this paragraph would be best if placed in the methods section. Although the revised introduction adequately addresses my other comments I still feel that a description if the specific gradient forest methods here is out of place. Although these methods are certainly key to the analytical structure of the assessment, noting them in the introduction feels abrupt, especially after the paragraph noting the hypothesis examined but before the description of study area. Given that the gradient forest methods aren’t the key focus of this study, rather environmental drivers of cryptic coral composition, the introduction would benefit from a reduced focus on the particular methods.

>>>Response – We moved this paragraph to the Methods (now lines 256-267).

Supplementary Figures: These appear to be out of order in the main text. Although this will probably be addressed in copy-editing, it may be best to rearrange them at this stage.

>>>Response – We re-ordered all supplementary figures and tables to be consistent with their order in the main text.

Semivariograms and kriging approach:

I appreciate the authors providing these supplementary figures—they greatly enhance the interpretation of the spatial interpolation of environmental conditions. Although, it is unclear what the numbers next to each point represent. Also, shouldn’t the points be pairwise comparisons of environmental stations? If so, shouldn’t there be many more points for each plot which may improve model fit.

>>>Response – Data points are grouped into bins based on their lag distance, so that all pairs of points with similar distances are analyzed together. Thus, each point in the variogram plot represents the average semivariance for all data pairs within a bin. The label next to each point indicates the number of data pairs used to calculate the semi-variance at each distance bin.

We added this explanation along with the following text to clarify the variogram plots in the supplementary figures:

Supplementary Figure 3: “Semi-variograms are included to the right of each map, which depict spatial autocorrelation between environmental measurements across the seascape. The x-axis of the variogram plot refers to the “lag distance,” or the distance between pairs of data points (expressed here in degrees of latitude and longitude). Data points are grouped into bins based on their lag distance, so that all pairs of points with similar distances are analyzed together. Thus, each point in the variogram plot represents the average semivariance for all data pairs within a bin. By default, the R package automap ensures that each bin contains at least five data pairs; if any bin has fewer than five pairs, it is merged with the adjacent bin. The y-axis represents the calculated semi-variance at each distance bin. Finally, the label next to each point indicates the number of data pairs used to calculate the semi-variance at each distance bin.

From the resulting variogram plots, several parameters controlling the fit of the model can be interpreted- including the “nugget,” or the y-intercept of the variogram, which represents small-scale variability of the data likely attributed to measurement error. The “range” is the distance where the variogram levels off (i.e., distance where spatial autocorrelation reduces), and the “sill” is the variance where the variogram levels off. 

When interpreting the variogram, lag distance increases along the x-axis. For certain variables, such as mean temperature, the semivariance on the y-axis generally increases, indicating a reduction in spatial correlation between data points as they become further apart. For other variables, such as minimum temperature, semivariance appears relatively constant across lag distances, implying a more uniform spatial distribution of values through space.”

As the authors note, the fit of these models is quite variable: some perform better than others. Based on even the good fits, I am somewhat concerned about the spatial scale of interpolation. For example, the fit for DO_max isn’t very good over the distance of ~0.03 (what are the units here?). For other variables (e.g., monthly range in Nitrogen), the fit line doesn’t appear to fit the [negative] trend at all. Given these fits, I am concerned that the environmental patterns over space may not be adequately represented by the models described. In general, some of the fits appear to either (1) obscure the spatial variability inherent to some variables or (2) don't appropriately capture trends.

>>>Response – We agree that the environmental patterns are a critical factor and have taken the following steps to address your concerns- 

To improve the fit of each variogram, we re-interpolated the variables individually, allowing us to visually inspect and refine the model fit for each case rather than relying solely on an automated kriging procedure. This iterative approach enabled us to ensure better alignment between the variogram models and the observed data. The updated interpolated variables are presented in new Supplementary Figure 3 (included in a separate PDF attachment to preserve figure resolution).

While we believe these adjustments have generally improved the overall fit of the models, we acknowledge that some variograms still exhibit relatively high variability, particularly at smaller lag distances. We provided additional explanation (see pasted lines below), justifying why we consider this variability acceptable for our analysis. Furthermore, we note that repeated testing of various model fits did not substantially alter the overall spatial patterns, suggesting that the models are adequately capturing the larger-scale environmental trends despite local inconsistencies.

Supplementary Figure 3: “Regarding the fit of the variogram model, we note that the initial points in the plot often exhibit high variability due to the small lag distances between data pairs. These points are calculated from fewer data pairs, as the number of available pairs tends to increase with larger distances. Moreover, the high variability among the initial points is often an artifact of observations being sparse or clustered around certain regions.

For example, in our study, the high variability at small lag distances in all variables is most likely due to more frequent sampling near the capital, Christiansted (see Supplementary Fig. 1), where environmental monitoring efforts are more concentrated. These localized measurements contribute disproportionately to the small lag distance bins. To minimize the impact of this artifact, we fit the variogram model to the remaining data points, excluding the early, high-variability points.

Some variogram models show a better fit than others (ex. mean nitrogen, mean dissolved oxygen, and mean temperature), thus the resulting interpolations for those variables are more likely to capture actual trends across the seascape. On the other hand, models with a poorer fit can still identify regions with exceptionally high values (ex. mean E. coli) or low values (ex. monthly range of Nitrogen). However, we opted to retain even these gradients as predictors to see if those spatial patterns correlated with genetic structure. If variables with weaker variograms demonstrated high associations with genetic structure, we would have interpreted that result as genetic turnover across the seascape without attributing it to our interpolated variables, as even well-predicted variables would need to be validated experimentally. We did not, however, retain variables without variability across the seascape (ex. minimum nitrogen) and removed those variables as predictors from gradient forest models.”

Regarding your concern about the units for DO, we added clarification for units of all variables in the manuscript:

Lines 210-211: “Variables included pH, Enterococcus and E. coli (count per 100ml), dissolved oxygen, Kjeldahl nitrogen, and phosphorus (mg/L), and Secchi disk depth (m).”

Following the re-interpolation of our environmental variables, we also redefined the ecoregions and re-ran the gradient forest model using the updated set of variables. The resulting ecoregion patterns were consistent with those derived from the previous variables, and we have updated Figure 2 accordingly to reflect this.

In terms of model outcomes, after re-running the gradient forest analysis, we found that depth remained by far the most important variable, followed by pH_yearly_range as the second most influential factor, followed by temperature-related variables (see new Figure 5). It is worth noting that the importance of pH_yearly_range is consistent with our initial manuscript submission, where it was also identified as the second most important variable.

Reviewer #2: 

1) Overall I thought the revised text was fairly superficial, but it more or less addresses the point. For example, the phrase “stepped clines driving tension zone formation and assortative mating” is quite jargony and “other factors, such as ocean currents influencing larval dispersal” is quite vague. The statement that “the ability of currents to sustain high dispersal and connectivity” (L462) seems at odds with all the research showing that larvae (including corals) disperse much less than expected from ocean currents. For example: Prata et al 2024 Some reef-building corals only disperse metres per generation. Proc. R. Soc. B 291: 20231988. https://doi.org/10.1098/rspb.2023.1988

>>>Response – We omitted these vague and jargony sentences, and we added the following new paragraphs to our discussion that consider alternative explanations for cryptic community structure. These paragraphs focus on possible non-environmental drivers that could have influenced the patterns we observed but were beyond the scope of our current analysis. Specifically, we discuss factors such as ocean currents, reproductive strategies (e.g., brooding vs. spawning), natural disturbances, and prezygotic barriers, including gamete incompatibility and temporal isolation due to differences in spawn timing.

Regarding your comment on ocean currents and larval dispersal, we acknowledge that Prata et al. 2024 highlights the limited dispersal distances of reef-building corals. We revised our discussion point to clarify that while ocean currents may influence connectivity, the actual dispersal distance for some species could be much shorter.

Lines 456-511: “While we can estimate the extent of environmental differences that contribute to community structure, we recognize that most lineages are not strictly confined to specific depths or ecoregions and can co-occur at certain sites (see Fig 4). Untangling the mechanisms driving and maintaining genetic differentiation between cryptic lineages in the absence of geographic isolation remains a challenge. In addition to environmental variables, non-environmental factors such as ocean currents, reproductive strategies, natural disturbances, and prezygotic barriers may also influence coral community structure. For example, ocean currents can restrict dispersal patterns of local coral taxa (Fiesinger et al., 2023, Wood et al., 2016) and determine the settling locations of coral larvae (Thompson et al., 2018). Seasonal variation in the strength and direction of currents can further affect larval connectivity (Gilmour et al. 2016). In St. Croix, currents bend around the southern coast, sending wake flows to both the eastern and western sites (Chérubin & Garavelli, 2016), which may help explain the structured distribution of cryptic lineages across the east and west. Additionally, the reproductive mode of coral species- whether brooding or spawning- influences dispersal distance due to variations in larval phase length. For instance, the broadcast spawner Acropora palmata can have parent-offspring separations of 70 meters to one kilometer (Aurélien et al.), while brooding Agaricia corals typically only disperse 2 to 11 meters per generation (Prata et al. 2024). Despite the influence of ocean currents, many coral larvae settle in close proximity to their parent colonies (Prata et al. 2024), suggesting that short dispersal distances may limit the role of currents in driving genetic differentiation within these populations.

Natural disturbances, such as hurricanes, can also shape the structure of cryptic coral communities by fragmenting corals and redispersing them over large distances. However, the impact of these disturbances varies depending on oceanic geographic features. For example, patterns of reef destruction in St. Croix caused by Hurricane Hugo in 1989 varied by depth, shoreline orientation, and the composition of benthic communities before the storm (Hubbard et al. 1991). Moreover, patterns of coral growth on the mesophotic shelf edge of the U.S. Virgin Islands appear to be structured by acute but infrequent swell impacts, which varies across depths (Smith et al. 2016). These examples illustrate how natural disturbances can generate distinct patterns in coral community structure, shaped by the interaction between damage and recovery processes. On St. Croix, the impacts of Hurricanes Irma and Maria were similarly devastating, with significant damage to heritage and fisheries resources (Dunnavant et al. 2018; Stoffle et al. 2020), though the full scope of damage to the island’s coral communities remains unclear.

Prezygotic barriers, such as temporal isolation and gamete incompatibility, can also drive genetic divergence and the formation of cryptic lineages in corals. Within a single species, individuals may spawn at different times of the year, with some spawning in spring and others in fall. This asynchronous spawning is observed in species such as Acropora tenuis, A. samoensis, A. digitifera, Orbicella spp., and Mycedium elephantotus, all of which show genetic differentiation linked to their spawning periods (Dai et al., 2000; Gilmour et al., 2016; Levitan et al., 2011; Ohki et al., 2015; Rosser, 2015). Spawn timing is also determined by environmental cues and genetics, and even just a few hours difference in spawning can lead to sympatric speciation (Monteiro et al. 2012). Gamete incompatibility further contributes to this process. In some Orbicella spp., eggs can demonstrate conspecific sperm precedence (CSP), where they preferentially accept sperm from their own species. CSP may help gametes from broadcast spawners find each other in the water column, but it may also drive divergence in gamete compatibility (Fogarty et al. 2012). In the Caribbean, three recently diverged Orbicella spp.- O. franksi, O. faveolata, and O. annularis- show varying levels of gamete incompatibility, which may have played a role in their speciation (Levitan et al. 2004). 

Despite these alternative explanations for the observed distribution patterns, depth consistently appears to be a signi

---

## [Editor Report · Decision Letter 2]

21 Jan 2025

Cryptic coral community composition across environmental gradients

PONE-D-24-30961R2

Dear Dr. Black,

We’re pleased to inform you that your manuscript has been judged scientifically suitable for publication and will be formally accepted for publication once it meets all outstanding technical requirements.

Kind regards,

Vitor Hugo Rodrigues Paiva, Ph.D.

Academic Editor

PLOS ONE
---

## [Editor Report · Acceptance letter]

24 Jan 2025

PONE-D-24-30961R2 

PLOS ONE

Dear Dr. Black, 

I'm pleased to inform you that your manuscript has been deemed suitable for publication in PLOS ONE. Congratulations! Your manuscript is now being handed over to our production team.

Kind regards, 

on behalf of

Dr. Vitor Hugo Rodrigues Paiva 

Academic Editor

PLOS ONE